# Developmentally Regulated CYP2E1 Expression Is Associated with a Fetal Pulmonary Transcriptional Response to Maternal Acetaminophen Exposure

**DOI:** 10.3390/biomedicines13102446

**Published:** 2025-10-08

**Authors:** Emma M. Golden, Zhuowei Li, Lijun Zheng, Mack Solar, Maya R. Grayck, Nicole Talaba, David J. McCulley, David J. Orlicky, Clyde J. Wright

**Affiliations:** 1Section of Neonatology, Department of Pediatrics, University of Colorado School of Medicine, Aurora, CO 80045, USA; 2Division of Neonatology, Department of Pediatrics, University of California, San Diego, CA 94143, USAdmcculley@health.ucsd.edu (D.J.M.); 3Department of Pathology, University of Colorado Anschutz School of Medicine, Aurora, CO 80045, USA; david.orlicky@cuanschutz.edu

**Keywords:** acetaminophen, CYP2E1, prenatal, fetal, lung injury, paracetamol

## Abstract

**Background/Objectives**: Acetaminophen (APAP) is used during 50–60% of pregnancies in the U.S. and has been associated with childhood respiratory morbidity, though the underlying mechanism remains unclear. APAP-induced injury is dependent on cell-specific expression of CYP2E1, the enzyme that metabolizes APAP into the mitochondrial toxin NAPQI. In mice, pulmonary *Cyp2e1* expression peaks during the saccular stage of lung development on embryonic day 18 (E18). We investigated whether this developmental surge in *Cyp2e1* triggers a pulmonary transcriptional response to maternal APAP exposure in embryonic lungs. **Methods**: Pregnant dams were exposed to APAP on E17 or E18 (150 or 250 mg/kg, IP) using doses derived from prior studies. We assessed the induction of NRF2 target genes and genes associated with inflammation, apoptosis and cellular stress due to their roles in APAP-induced oxidative and cellular stress. **Results**: At E17, maternal treatment with APAP induced pulmonary *Cyp2e1* but resulted in inconsistent transcriptional changes. In contrast, maternal APAP at E18 triggered a robust transcriptional induction of *Cyp2e1*, NRF2 targets and markers of apoptosis, inflammation and cellular stress. Histopathology at birth after E18 APAP exposure revealed no acute pulmonary injury. **Conclusions**: We demonstrate a developmentally regulated, dose-dependent transcriptional response to maternal APAP in the embryonic murine lung. Importantly, transcriptional responses do not directly indicate lung injury; thus, future studies should assess protein-level changes following APAP exposure. This study underscores the need for further investigation into the role of developmentally regulated *Cyp2e1* expression in APAP-induced toxicity and long-term respiratory morbidity.

## 1. Introduction

Acetaminophen (APAP) is the most commonly used analgesic and antipyretic around the world, with exposures occurring at least once in 50–60% of pregnancies in the United States and Europe [1,2,3,4,5]. Common indications for APAP use during pregnancy include chronic pain, headaches, injury, illness and sleep difficulties [6,7,8]. Notably, over two thirds of reported APAP use is self-directed and without physician oversight, making it difficult to determine how frequent and how much APAP women often take during pregnancy [7,9,10]. Two main factors for the use of APAP during pregnancy include the desire to reduce opioid exposure and the relatively ‘fair’ safety profile of APAP. Historically, there has been a general recommendation by providers that APAP use during pregnancy could be considered risk-free for both the mother and the fetus when taken at recommended doses [2,11,12]. Recently, the Food and Drug Administration (FDA) initiated a safety label change and notified physicians of a possible link between APAP use during pregnancy and subsequent diagnosis of conditions like autism and ADHD. However, they state that the choice still belongs to parents, and APAP remains readily available for pregnant women to use [13].

Despite a presumed low risk with fetal exposure, several studies and meta-analyses have linked prenatal and perinatal APAP exposures following maternal APAP intake to childhood respiratory morbidity [14,15,16,17,18,19,20,21,22,23,24,25]. Notably, childhood asthma is one of the most commonly implicated lung morbidities that has been associated with APAP use. This association has historically been disregarded due to the lack of cellular and molecular mechanisms linking APAP use to lung injury. Proposed mechanisms include oxidative/inflammatory injury induced through glutathione consumption, a Th1/Th2 imbalance favoring the Th2 allergic phenotype and IgE-mediated disease [19,26,27,28]. However, previous studies fail to demonstrate a consistent relationship between APAP exposures and allergen sensitization [29,30] and IgE levels [31,32,33,34], even when the relationship between APAP and asthma is found. A limitation in these studies that must be considered is the variable phenotype leading to the diagnosis of asthma in early childhood. The majority of these studies categorize the diagnosis of childhood asthma based on parental reports, presence of a prescription for asthma medications, or chart reviews documenting a provider’s diagnosis [21,23,25,29,30,32,33,34,35,36,37,38]. Few studies rely on pulmonary function as a basis for diagnosis, which to this day serves as the primary test used for diagnosing asthma in children [39]. In the absence of a definitive link between APAP exposure and asthma/allergic phenotype, alternative mechanisms linking exposure and abnormal lung development and function should be considered.

In nearly all adult mammals, APAP is metabolized primarily in the liver into toxic and non-toxic metabolites [40,41]. The majority of hepatocyte metabolism in the adult liver occurs via glucuronidation or sulfation [42,43,44]. Additionally, a small percentage of APAP is metabolized to the toxic metabolite *N*-acetyl-*p*-benzoquinone imine (NAPQI) by the xenobiotic enzyme cytochrome P450 family 2 subfamily E member 1 (CYP2E1) primarily in the centrilobular hepatocytes. Rapid detoxification of NAPQI occurs via glutathione (GSH) conjugation. In the presence of alternative metabolizing pathways (glucuronidation and sulfation) and detoxifying factors (glutathione), APAP is safe when taken at recommended doses. However, with toxic APAP exposure, NAPQI accumulates, leading to the depletion of hepatic glutathione, which in turn leads to its subsequent covalent bonding with intracellular proteins. The APAP-protein adduct formation within the mitochondria contributes to cellular dysfunction and ultimately results in the activation of death signaling pathways in hepatocytes and consequently hepatocellular necrosis [45,46,47]. Notably, the relatively high cell-type-specific expression of *Cyp2e1* in the centrilobular hepatocyte of the mature liver explains why hepatotoxicity is the most common cause of death following overdose [42,48].

CYP2E1 expression is not limited to the liver. Pulmonary *Cyp2e1* expression has been reported in both the adult human and rodent lung [49,50,51,52,53,54,55,56,57,58]. Emerging data indicate that pulmonary *Cyp2e1* expression is both dynamic and specific to cell types. In the murine lung, *Cyp2e1* follows a biphasic developmental pattern, with expression surges at the beginning of the saccular stage of lung development around embryonic day 18 (E18) and again during the second phase of murine lung alveologenesis at postnatal day 14 (P14) [59,60,61,62,63,64,65]. These developmentally regulated peaks may represent critical windows of susceptibility to APAP-induced lung injury, although this has not yet been established.

Whether direct pulmonary toxicity secondary to CYP2E1 expression explains the reported association between prenatal APAP exposure and early childhood respiratory morbidity is unknown. Experimental data on maternal APAP exposures and fetal pulmonary outcomes are limited, with limited pre-clinical studies addressing the question [66,67]. Both studies focused primarily on airway inflammation and IgE-mediated disease as potential mechanisms, without exploring CYP2E1-dependent toxicity or assessing key developmental timepoints.

Therefore, we set out to investigate the developmentally regulated pulmonary expression of *Cyp2e1* and assess its potential association with changes in the fetal pulmonary response to maternal APAP exposure. We found that a surge of *Cyp2e1* expression occurs at E18, coinciding with the onset of the saccular stage of development, in both female and male mice. We also identified a developmentally regulated and dose-dependent transcriptional response in the fetal murine lung following maternal APAP exposure. This response encompassed activation of NRF2 target genes as well as genes related to apoptosis, inflammation and cellular stress. Our preclinical data implicate a biological mechanism linking prenatal APAP exposure with childhood pulmonary morbidity, underscoring the need for further studies to establish the clinical relevance of these findings.

## 2. Materials and Methods

### 2.1. Murine Model of Acetaminophen Exposure

All procedures were approved by the IACUC at the University of Colorado (Aurora, CO) and in an American Association for Accreditation of Laboratory Animal Care—accredited laboratory animal facility. The treatment, care and handling of the animals was in accordance with the National Institutes of Health guidelines for ethical animal treatment.

To evaluate transcriptional responses, C57BL/6 murine dams were exposed to a single dose of APAP [either 150 or 250 mg/kg, intraperitoneal (IP); dissolved in phosphate-buffering saline (Corning, Manassas, VA, USA) on embryonic day 17 (E17) or 18 (E18) during the early phases of murine lung saccular stage of development when pulmonary *Cyp2e1* expression surges. Mice were sacrificed 6 h after exposure, and tissue samples were collected as previously described [68].

To evaluate histopathologic data, C57BL/6 murine dams were exposed to APAP (250 mg/kg IP, ×1 dose) at E18 and were allowed to deliver.

### 2.2. mRNA Extraction and Quantitative Real-Time PCR

For ontogeny, lungs collected from E12 to P21 mice were homogenized in TRIzol (Thermo Fisher Scientific, Waltham, MA, USA), and RNA was isolated using the RNeasy Plus Mini Kit (QIAGEN, Boston, MA, USA). cDNA was made using the Superscript III First-Strand Synthesis System (Invitrogen, Carlsbad, CA, USA; Thermo Fisher Scientific, Waltham, MA, USA). The expression timeline of Cyp2e1 in the lung was presented as relative gene expression, normalized to β-actin. Each timepoint was the average of three independent samples. qRT-PCR was quantified using the SYBR green (Applied Biosystems, Waltham, MA, USA; Bio-Rad, Hercules, CA, USA) and run on a LightCycler 480 (Roche, Belmont, CA, USA) and a CFX Connect System (Bio-Rad, Hercules, CA, USA).

For evaluating acute transcriptional response, fetal lungs were removed, snap-frozen in liquid nitrogen and stored at −80 °C. Frozen tissue was placed in RLT-βME buffer (Qiagen, Valencia, CA, USA), and tissue was homogenized using the Bullet Blender (NextAdvance, Troy, NY, USA). Pulmonary mRNA was collected using the RNeasy Mini Kit (Qiagen, Valencia, CA, USA), assessed for purity/concentration using the NanoDrop (ThermoFisher Scientific, Waltham, MA, USA) and converted into cDNA using the Verso cDNA synthesis Kit (Thermo Scientific, Waltham, MA, USA). Relative mRNA levels were evaluated by quantitative real-time PCR using the TaqMan gene expression system (Applied Biosystems, Foster City, CA, USA). Gene expression was assessed with predesigned exon-spanning primers using the StepOnePlus Real-Time PCR System (Applied Biosystems, Foster City, CA, USA) (Table 1). Quantification was performed using the cycle threshold (ΔΔCt) method using the housekeeping gene *18S*. Primers used can be found in Table 1.

### 2.3. Lung Inflation and Collection of Pulmonary Tissue

For pulmonary mRNA and protein analysis, E18 pups were delivered and sacrificed after 6 h of APAP exposure with a fatal dose of pentobarbital sodium. Lungs were perfused with normal saline, removed, snap-frozen and stored at −80 °C. For morphometric assessments, murine dams were exposed to APAP on E18 and were allowed to deliver. After delivery, dams and pups were sacrificed with a fatal dose of pentobarbital sodium. Following perfusion of the lungs with normal saline, the trachea was cannulated with a 24 G angiocath, and the lungs were inflation-fixed at 25 cm H_2_O pressure for 10 min with 4% paraformaldehyde. Lungs were paraffin-embedded, and sections were cut (5 µm) and stained with hematoxylin and eosin at the University of Colorado Anschutz Medical Campus Pathology Shared Resource Core.

### 2.4. Histologic Evaluation of APAP-Induced Pulmonary Injury

Histopathological scoring of the lungs was performed by a trained histologist blinded to the treatments or grouping of animals as previously reported [69]. Briefly, 8 semi-quantitative criteria were used for this scoring: (1) the integrity of the respiratory and terminal bronchiole epithelium (0–3, normal to severe, as well as the presence or no presence of apoptotic epithelium in the airway lumen); (2) the relative quantity of bronchus-associated lymphoid tissue (BALT; 0–2, none to lots); (3) the peripheral airway macrophage load (0–3, none to lots with clumps); (4) the presence of peripheral lung emphysema with alveolar wall clubbing (0–2, none to lots); (5) the relative quantity of peri-bronchial polymorphonuclear leukocytes (PMNs) (0–2, none to lots); (6) the relative quantity of peri-vascular PMNs (0–2, none to lots); (7) epithelial blebs (0–2, none- lots); (8) relative thickness of the alveoli septae (1–3, mild to severe). Scores were tallied to create a “Total Injury” score. Histological images were captured on an Olympus BX51 microscope equipped with a 17mp Olympus DP73 high-definition, color, digital camera using the Olympus CellSens software V4.4 (Olympus, Waltham, MA, USA). All composite images were cropped and assembled using Adobe Photoshop Version 26.9.

### 2.5. Statistical Analysis

For comparison between treatment groups, the null hypothesis that no difference existed between treatment means was tested by Student’s *t* test for two groups and two-way ANOVA for multiple groups with potentially interacting variables (time, APAP exposure), with statistical significance determined by means of the Bonferroni method of multiple comparisons (Prism 10.0, GraphPad Software, Inc., Boston, MA, USA). Statistical significance was defined as *p* < 0.05.

## 3. Results

### 3.1. Pulmonary CYP2E1 Expression Peaks at E18 and Exceeds E17 Levels

Public murine data available from LungMAP demonstrates a prenatal surge in pulmonary *Cyp2e1* in the saccular stage of lung development [59,60,61,62,63,64]. We sought to confirm these findings and study sex-specific expression. Using whole lungs of wild-type mice at various prenatal and postnatal timepoints, we found that *Cyp2e1* expression is significantly increased compared to expression at E16 and P0 (Figure 1A). This timing is consistent with the beginning of the saccular stage of lung development [65]. To investigate the peak of expression at E18, we evaluated *Cyp2e1* between E17 and E18. In both females and males, dynamic increases in pulmonary *Cyp2e1* expression were noted between E17 and E18 (Figure 1B,C). Pulmonary expression at E17 (Figure 1D) and E18 (Figure 1E) was similar between male and female mice.

### 3.2. Acetaminophen Exposure Does Not Result in Pup Mortality or Significant Weight Difference

Having noted a developmentally regulated prenatal peak in *Cyp2e1* expression, we interrogated for the presence of a pulmonary transcriptional response associated with this dynamic surge. We chose to investigate using two doses, 150 and 250 mg/kg IP. These dosages were chosen based on a thorough literature review of previous murine APAP exposures, which demonstrated that these two doses are nonfatal, with the 150 mg/kg dose being subtoxic and the 250 mg/kg dose demonstrating maternal hepatotoxicity [70,71,72,73,74]. These doses approximate human single exposure of 12 mg/kg (which is within the recommended single dosage) and 20 mg/kg (which is slightly higher than the recommended single dosage), respectively [69]. Therefore, we examined the effect on pups exposed to the 250 mg/kg dose. Consistent with previous reports [67], our results show that a single 250 mg/kg exposure resulted in APAP-induced liver damage in dams (Figure 2A).

To evaluate offspring outcomes following maternal exposures, dams treated with 250 mg/kg APAP at E18 were allowed to spontaneously deliver 7–71 h later. No pup mortality was observed, with an average litter size of 6.5 (range 6–8). Moreover, no significant differences in body weight were detected between female (Figure 2B) or male (Figure 2C) pups from PBS- and APAP-exposed dams (*p* > 0.05).

#### 3.2.1. Acetaminophen Exposure Induces Expression of *Cyp2e1* in E17 Female and Male Mice

Having determined the feasibility of exposing pregnant dams to APAP at both the 150 and 250 mg/kg doses at E17 and E18, we next interrogated the fetal pulmonary transcriptional response. Previously, we demonstrated that APAP exposure increases *Cyp2e1* expression in the P7, P14 and adult murine lung [64,75,76,77]. Thus, we interrogated whether the fetal pulmonary *Cyp2e1* expression increased in response to maternal APAP exposure. In this study, we found significant increases in pulmonary *Cyp2e1* expression in both female and male mice at E17 following maternal APAP exposure at both the 150 mg/kg and 250 mg/kg dosages (Figure 3A).

#### 3.2.2. Prenatal Acetaminophen Exposure Does Not Result in a Consistent Pattern of Acute Transcriptional Response in Markers of APAP-Induced Cellular Injury in E17 Female and Male Mice

To determine if fetal exposure to APAP results in transcriptional evidence of response to oxidative stress, we assessed the NRF2 (Nuclear erythroid 2 p45-related factor 2) target genes *Gclc*, *Nqo1* and *Hmox1*. NRF2 plays a central role in the induction of antioxidant gene transcription and protects the lung from oxidative stress [78,79,80,81,82]. We additionally used induction of inflammatory (*Mmp9* and *Il6*) and pro-apoptotic genes (*Trp53*, *Puma*, *Noxa*) as evidence of APAP-induced cellular injury [46,83,84,85,86,87]. Notably, these markers have been reported in hepatic tissues following APAP exposure but remain to be fully explored in the developing lung. Lastly, we interrogated the induction of genes involved in cellular stress responses. *Ckdn1a*, which encodes P21, acts as a crucial mediator in the acute cellular response in the lung after oxidative-stress-induced injury and has been associated with neonatal lung injury and senescence [88,89,90]. Following APAP administration, the lung has been shown to demonstrate evidence of endoplasmic reticulum (ER) stress in the adult murine models [76]. Using *Ddit3* and *Atf4*, which are known ER-stress-related marker genes, we investigated ER stress in prenatal APAP exposures [76,91]. While there were some statistically significant transcriptional changes observed at E17, neither dose elicited a consistent transcriptional change across NRF2 target genes or in markers of inflammation, apoptosis or cellular stress in either female or male mice at the timepoints investigated (Figure 3B–E).

### 3.3. Acetaminophen Exposure Induces Acute Transcriptional Response in E18 Female and Male Mice

At the 150 mg/kg dose, E18 lungs demonstrate attenuation of *Cyp2e1* expression in response to maternal APAP exposure (Figure 4A). In contrast, at E18, there is a significant increase in pulmonary *Cyp2e1* expression in both female and male mice following the 250 mg/kg IP dose (Figure 4A). Similarly, at the 150 mg/kg exposure, there is no consistent pattern of injury in NRF2 target genes (*Gclc*, *Nqo1*, *Hmox1*), inflammation (*Mmp9* and *Il6*), apoptosis (*Trp53*, *Puma*, *Noxa*) and cellular stress response elements (*Cdkn1a*, *Ddit3*, *Atf4*) (Figure 4B–E). In contrast, at E18, exposed mice exhibited evidence of a transcriptional response to oxidative stress (*Gclc*, *Nqo1*, *Hmox1*) and cellular stress (*Cdkn1a*, *Ddit3*, *Atf4*) responses with the 250 mg/kg dose. This was accompanied by increased expression of the inflammatory marker *Mmp9* and the pro-apoptotic *Noxa* in both sexes, as well as *Puma* in females only (Figure 4B–E).

### 3.4. Acetaminophen Exposed (250 mg/kg IP × 1) Does Not Induce Histologic Evidence of Acute Pulmonary Injury in E18 Female or Male Mice

As previously described, we exposed a cohort of dams at E18 to either PBS control or APAP (250 mg/kg IP × 1 dose) and allowed them to deliver between 7 and 71 h post exposure. Blinded histopathologic evaluation of exposed fetal lungs was performed. There was no consistent pattern of pulmonary injury in exposed female or male mice (Figure 5A). Objective scoring in E18 female and male mice exposed to APAP was performed (Figure 5B–S). In all categories, there was no significant difference noted between PBS control and APAP-exposed female and male mice.

## 4. Discussion

We found that a single maternal APAP exposure (250 mg/kg IP) late in gestation at E18 induces an acute transcriptional response in the developing murine lung. This transcriptional response correlates with a dynamic change in pulmonary *Cyp2e1* expression, peaking at E18, the onset of the saccular stage of development, in both female and male mice. By closely evaluating the transcriptional response to APAP at two different doses (150 mg/kg and 250 mg/kg) across two developmental timepoints—embryonic day 17 (E17) and day 18 (E18)—we identified a window of transcriptional susceptibility in the developing fetal lung to maternal APAP exposure. The transcriptional response at this timepoint and dose includes upregulation of NRF2 target genes, proinflammatory mediators, apoptotic genes and key regulators of cellular stress response; all of which are consistent with exposure to oxidative stress and the known mechanisms of APAP toxicity [76,78,79,80,81,82,83,84,85,86,87,92]. While no histological evidence of acute injury was observed in the lungs of newborn mice exposed to APAP (250 mg/kg) at E18, whether this exposure and transcriptional response result in long-term functional and structural abnormalities is unknown.

Our findings are notable because they support our hypothesis that the appropriate study of the pulmonary implications of the in utero APAP exposures must consider developmentally regulated surges in *Cyp2e1* expression. Cellular toxicity following APAP exposures depends on cell-type-specific expression of *Cyp2e1* and accumulation of toxic metabolites [42,43,44]. Importantly, it has previously been reported that *Cyp2e1* expression peaks in the E18 murine lung [61,62]. In this project, we confirmed that this developmental surge in pulmonary expression is abrupt, occurring between E17 and E18, and occurs in both sexes. Additionally, this is the first study demonstrating that this developmentally regulated surge of pulmonary *Cyp2e1* expression is associated with an acute transcriptional response to an in utero APAP exposure.

Pulmonary *Cyp2e1* expression has been reported in both the adult human and rodent lung [49,50,51,52,53,54,55,56,57,58]. Studies have found expressions in the proximal and distal pulmonary epithelium, as well as cell-type-specific expressions in alveolar macrophages and club cells [51,53,54,93,94,95]. In addition, pulmonary *Cyp2e1* expression is inducible with APAP exposures [64,75,76,77]. APAP toxicity is dependent on the expression of *Cyp2e1*; therefore, it is not surprising that APAP-induced lung injury consistently occurs at hepatotoxic APAP exposures in rodents [95,96,97,98,99]. Preclinical studies have identified several key features of APAP-induced lung injury, including APAP-protein adducts detected in the lung following exposure [95,100,101,102,103], glutathione depletion and cell death [97,104,105,106]. Notably, preclinical studies show that APAP-induced lung injury occurs even in the absence of hepatic insult [75,96,101]. Consistent with these preclinical findings, clinical studies have associated acute APAP overdose with pulmonary injury in adults [107,108,109,110]. It is important to note that these data are limited to the adult lung and have not been similarly investigated in the in utero-exposed developing lung.

Few efforts have directly investigated the impact of APAP exposures in murine dams on fetal pulmonary outcomes. One early study evaluated chronic perinatal paracetamol exposures in murine dams and subsequent allergic airway disease in offspring early on. Dams were exposed daily to paracetamol via oral gavage (120 mg/5 mLs) from the day of mating throughout pregnancy, and pups were later challenged with allergens to assess airway hyper-responsiveness, inflammation and remodeling. This study concluded that chronic maternal exposure did not alter allergic airway disease outcomes or IgE levels in offspring [66]. Another study interrogated prenatal APAP exposures and pup airway inflammation [67]. This group exposed murine dams to a single APAP dose (50 mg/kg or 250 mg/kg, IP) at gestational day (gd) 12.5 and later examined adult offspring for eosinophilic airway inflammation. They reported increased airway inflammation in APAP-exposed offspring, characterized by leukocyte infiltration and tissue remodeling. While these studies pioneered investigating maternal APAP exposures and their effects on the developing lung, they focused largely on inflammation and proposed IgE-mediated mechanisms of injury, without considering the developmental regulation of APAP metabolism in the fetal lung. We were able to use these prior studies to determine non-toxic APAP doses. The rationale for exposure at gd12.5, as used by Karimi et al. [67], is unclear, particularly as our analysis of *Cyp2e1* expression demonstrates that fetal pulmonary expression at this timepoint is low, suggesting limited susceptibility to APAP-induced toxicity during this stage of development. Thus, the absence of pulmonary injury observed at gd12.5 may reflect developmental timing rather than lack of a mechanistic effect. In contrast, our study provides evidence that maternal APAP exposure elicits an acute transcriptional response in the fetal lung, coinciding with developmentally regulated surges in *Cyp2e1*. Specifically, we demonstrate that fetal lungs mount transcriptional responses involving oxidative stress, inflammation, apoptosis and cellular stress when exposed to APAP at E18. This novel insight underscores the importance of developmental timing and transcriptional responses when assessing fetal susceptibilities to APAP-induced injury.

Previous studies show that there is a link between prenatal APAP exposures via maternal APAP intake and childhood respiratory morbidity [15,16,17,18,19,20,21,22,23,24,25]. Proposed mechanisms behind this association include oxidative/inflammatory injury induced through glutathione consumption, a Th1/Th2 imbalance favoring the Th2 allergic phenotype and IgE-mediated disease; however, the majority of these studies fail to demonstrate a consistent relationship between APAP exposure and the proposed mechanism [19,26,27,28,29,30,31,32,33]. It is well described that APAP-induced injury is dependent on cell-type-specific expression of CYP2E1. Importantly, recent data indicate the developing murine lung expresses *Cyp2e1* in a developmentally biphasic pattern, peaking at the saccular stage of development (E18) [59,60,61,62,63,64]. However, prior studies have failed to consider pulmonary *Cyp2e1* and its developmental surge as a potential mechanism of APAP-induced lung injury. This report confirms the dynamic developmental surge of pulmonary *Cyp2e1* and that it occurs in both female and male mice. Furthermore, we show that during this surge, the developing murine lung mounts an acute transcriptional response to maternal APAP exposure.

We speculate that developmentally regulated, cell-type-specific *Cyp2e1* expression dictates the fetal murine lung’s transcriptional response to in utero APAP exposure at E18. Data from the NIH-funded Molecular Atlas of Lung Development Program (LungMAP) [61] demonstrates that *Cyp2e1* expression peaks during the saccular stage and is restricted to mesodermally derived secondary crest myofibroblasts, a finding supported by other published single-cell analyses [59,60,111]. These cells are essential for forming alveolar septa and establishing gas exchange surfaces [112,113,114,115]. Loss or dysfunction of this cell population impairs alveolarization and disrupts lung maturation [112,113,116]. We propose that developmentally regulated, cell-type-specific *Cyp2e1* expression in these myofibroblasts increases susceptibility to APAP exposure, leading to oxidative, inflammatory and cellular stress responses. Even partial loss of this critical cell population may be sufficient to drive the transcriptional responses observed here. Future studies are needed to interrogate this hypothesis by examining cell-type-specific responses to in utero APAP exposure during critical stages of lung development.

Our study has several limitations. This study utilized C57BL/6 mouse models, recognizing significant differences that exist between rodents and humans with regard to lung development timing, drug metabolism and physiological responses. Therefore, there are limitations to directly applying results obtained from mouse models to humans. Our study shows that the developmentally regulated surge of *Cyp2e1* at E18 is associated with a more robust transcriptional response; however, gain and loss of function are necessary to mechanistically link this transcriptional response to pulmonary *Cyp2e1* expression. While we assessed the effects of APAP on the developing lung as a function of acute transcriptional response and histopathologic assessment, we did not examine the long-term impact of APAP exposure on lung structure or function. This is an important consideration, as a transcriptional response alone does not confirm lasting effects on lung development. Additionally, this study analyzed a single timepoint: 6 h post-APAP exposure. While an acute transcriptional response was observed at this time, it is likely that additional dynamic responses occur at both earlier and later stages, underscoring the need for future investigations across multiple timepoints. We observed attenuation of *Cyp2e1* expression on E18 with the 150 mg/kg dose; however, no consistent pattern of response was detected across all other target genes. The basis behind this variability remains unclear. Future studies should interrogate the degree of *Cyp2e1* induction at varying doses at this timepoint, the impact of multiple or repeated doses and the extent to which these changes influence downstream transcriptional responses. While we did not observe histological evidence of APAP-induced acute injury, it would be necessary to study the lung structure and function of exposed pups at later timepoints to interrogate for impaired lung development.

In this study, we investigated two doses, 150 mg/kg IP and 250 mg/kg IP, and the clinical relevance of these doses remains unclear. These dosages were chosen based on a thorough literature review of previous murine APAP exposures, which demonstrated that these two doses are nonfatal, with the 150 mg/kg dose being subtoxic and the 250 mg/kg dose demonstrating hepatotoxicity [67,70,71,72,73,74]. These APAP dosages match a dose of 12 mg/kg (which is within the recommended single dosage) and 20 mg/kg (which is slightly higher than the recommended single dosage) in humans [69]. These comparisons were obtained using guidelines of the US Department of Health and Human Services, FDA [117]. IP administration in experimental studies involving rodents has been shown to be a justifiable route for pharmacological studies due to its reliable delivery to the liver via the portal system [118]. While we know that IP-administered drugs are absorbed by peritoneal vessels and directly delivered to the portal system, one limitation is that this route does not account for small metabolic changes that may occur in the lumen of the GI tract in oral administration. It is important to note that both our study and prior reports have demonstrated hepatic injury in APAP-exposed murine dams at 250 mg/kg [67]. Thus, the fetal response we observed could in part reflect secondary effects of maternal response to APAP rather than a direct impact of APAP on the fetal lung. However, if maternal response at the 250 mg/kg dose were the primary driver, we would expect a similar impact on the embryonic lung with exposures at either E17 or E18. Further work is needed to evaluate a range of APAP doses, specifically at the critical E18 timepoint.

## 5. Conclusions

We conclude that, consistent with developmentally regulated surges in pulmonary *Cyp2e1* expression, the developing murine lung demonstrates an acute, dose-dependent transcriptional response to maternal APAP exposure, particularly at the onset of the saccular stage of development. These findings underscore the importance of accounting for dynamic developmental changes in pulmonary *Cyp2e1* as a potential mediator of APAP toxicity. This report additionally supports previous studies investigating the developmental regulation of murine pulmonary *Cyp2e1* expression. Notably, our study is the first to demonstrate that developmental surges in pulmonary *Cyp2e1* expression can mediate transcriptional responses to maternal APAP exposure. By integrating the temporal dynamics of pulmonary *Cyp2e1* expression, we provide new mechanistic insights into how the timing of exposure may mediate potential APAP-induced toxicity. These novel data are important given the high rate of APAP exposures during pregnancy in the US and globally [1,2,3,4,5], as well as the ongoing debate regarding the association between prenatal exposures and childhood respiratory morbidity [16,17,18,19,20,21,22,23,24,25,26,27,28,29,30,31,32,33,34,35,36,37,38]. Our data suggest that discrete developmental windows may confer heightened susceptibility to APAP-associated lung injury in mice. Whether similar windows of susceptibility exist in the developing human lungs is unknown, but if identified, such knowledge could inform clinical guidance regarding safer timing of APAP use during pregnancy. Continued pre-clinical and clinical investigations are needed to clarify the long-term consequences of prenatal APAP exposure on the developing lung and to determine whether human lungs are similarly susceptible to APAP-induced injury during critical developmental periods.

## Figures and Tables

**Figure 1 biomedicines-13-02446-f001:**
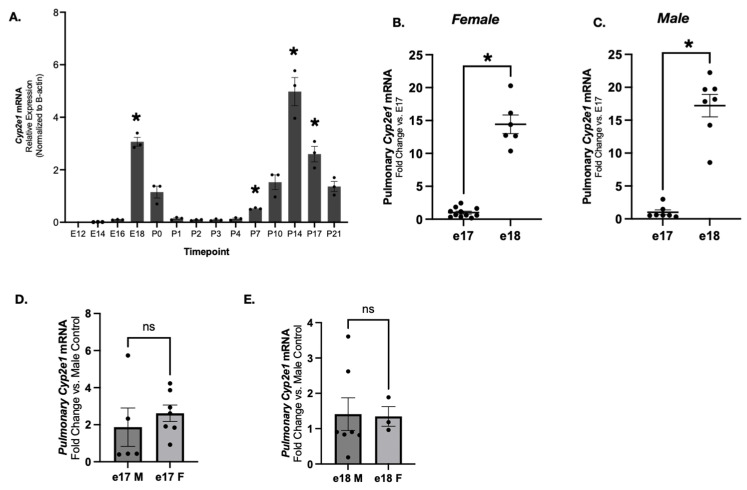
Pulmonary *Cyp2e1* expression peaks at the beginning of the saccular stage of lung development (E18). Pulmonary *Cyp2e1* expression at E18 exceeds E17 levels. Pulmonary *Cyp2e1* expression is similar between male and female mice at E17 and E18. Relative pulmonary *Cyp2e1* expression in whole lungs of wild-type mice at timepoints E12-P7 (**A**). * *p* < 0.05 vs. E12 control. Relative pulmonary *Cyp2e1* mRNA expression in E17 and E18 female (**B**) and male (**C**) in B6 mice. Data normalized to sex matched E17 expression. * *p* < 0.05 vs. E17 sex-matched control. Relative pulmonary *Cyp2e1* mRNA expression in E17 (**D**) and E18 (**E**) male and female mice. Data normalized to male age-matched expression. Data expressed as mean relative expression ± SEM. N = 9–16/age. ns, no statistical significance (*p* > 0.05).

**Figure 2 biomedicines-13-02446-f002:**
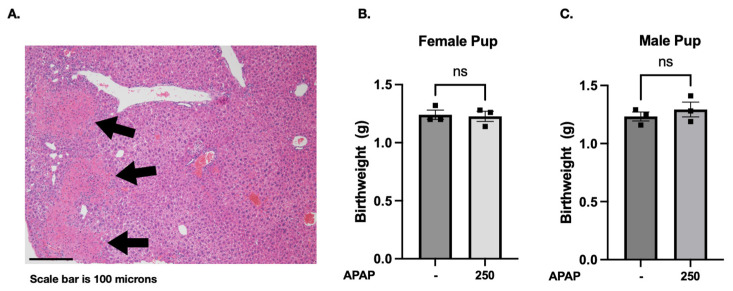
Acetaminophen exposure on E18 results in liver injury in pregnant mice. Acetaminophen exposure does not result in a significant weight difference. Liver histology of the murine dam following a single APAP exposure (250 mg/kg IP) (**A**). Arrows point to areas of necrosis. Average weight of pups per litter following PBS control versus APAP-exposed (250 mg/kg IP × 1 dose) on E18 in female (**B**) and male (**C**) pups. Pups weighed at time of birth, ranging 7–71 h following exposure. Litter size ranging 4–8 pups/litter. Statistical significance determined by *t* test between PBS control and APAP exposure. ns, no statistical significance (*p* > 0.05).

**Figure 3 biomedicines-13-02446-f003:**
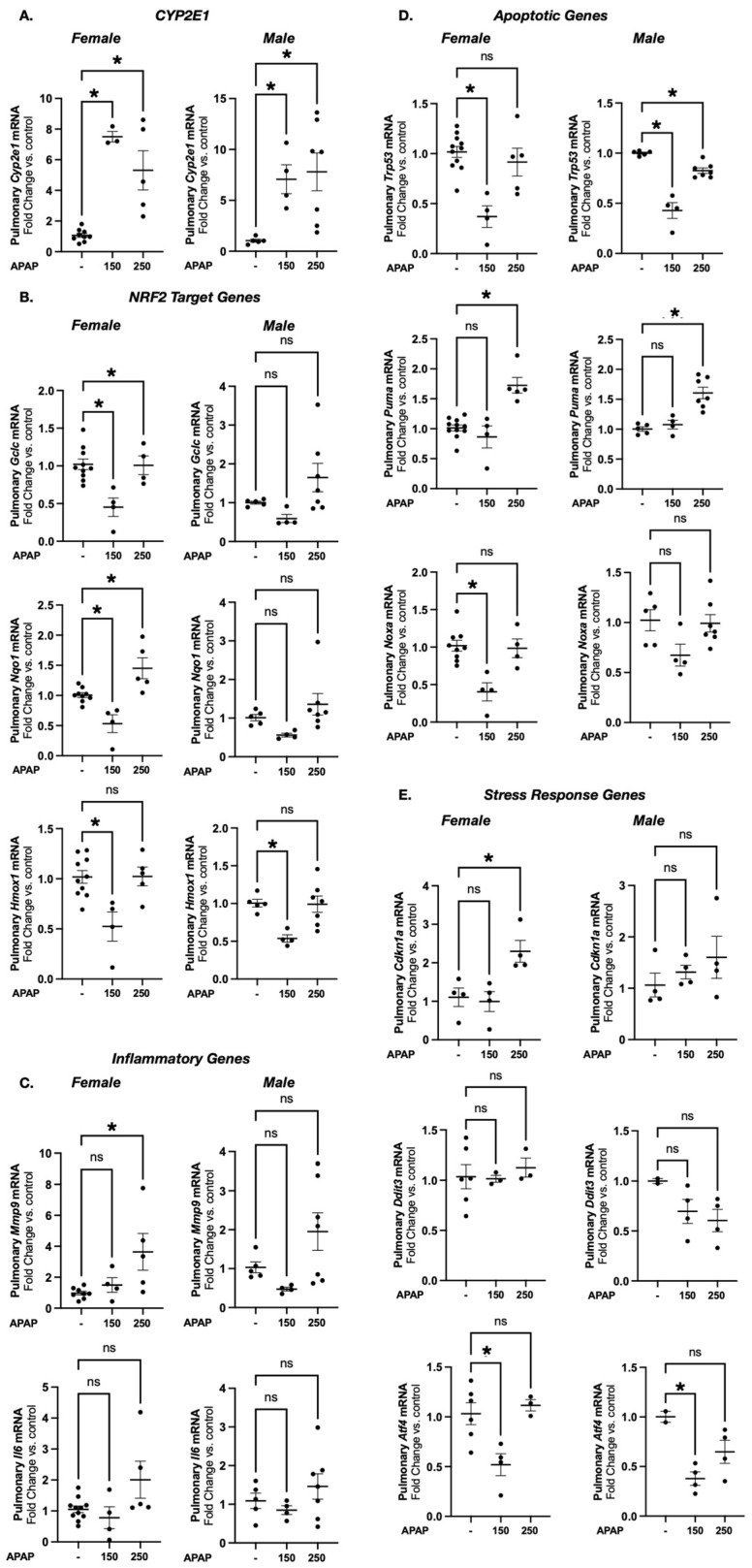
Prenatal acetaminophen exposure induces *Cyp2e1* expression in E17 female and male mice. Prenatal APAP exposure does not result in a consistent pattern of acute transcriptional response. Fold change in pulmonary mRNA expression of *Cyp2e1* (**A**); NRF2 target genes *Gclc*, *Nqo1*, *Hmox1* (**B**); inflammatory genes *Mmp9* and *Il6* (**C**); apoptotic genes *Trp53*, *Puma*, and *Nova* (**D**); and stress response genes *Cdkn1a*, *Ddit3*, *Atf4* (**E**) in APAP exposed (150 or 250 mg/kg IP × 1; 6 h) female and male E17 mice. Data expressed as mean relative expression ± SEM. N = 4–9/dose. * *p* < 0.05 vs. vehicle-exposed control. ns, no statistical significance (*p* > 0.05).

**Figure 4 biomedicines-13-02446-f004:**
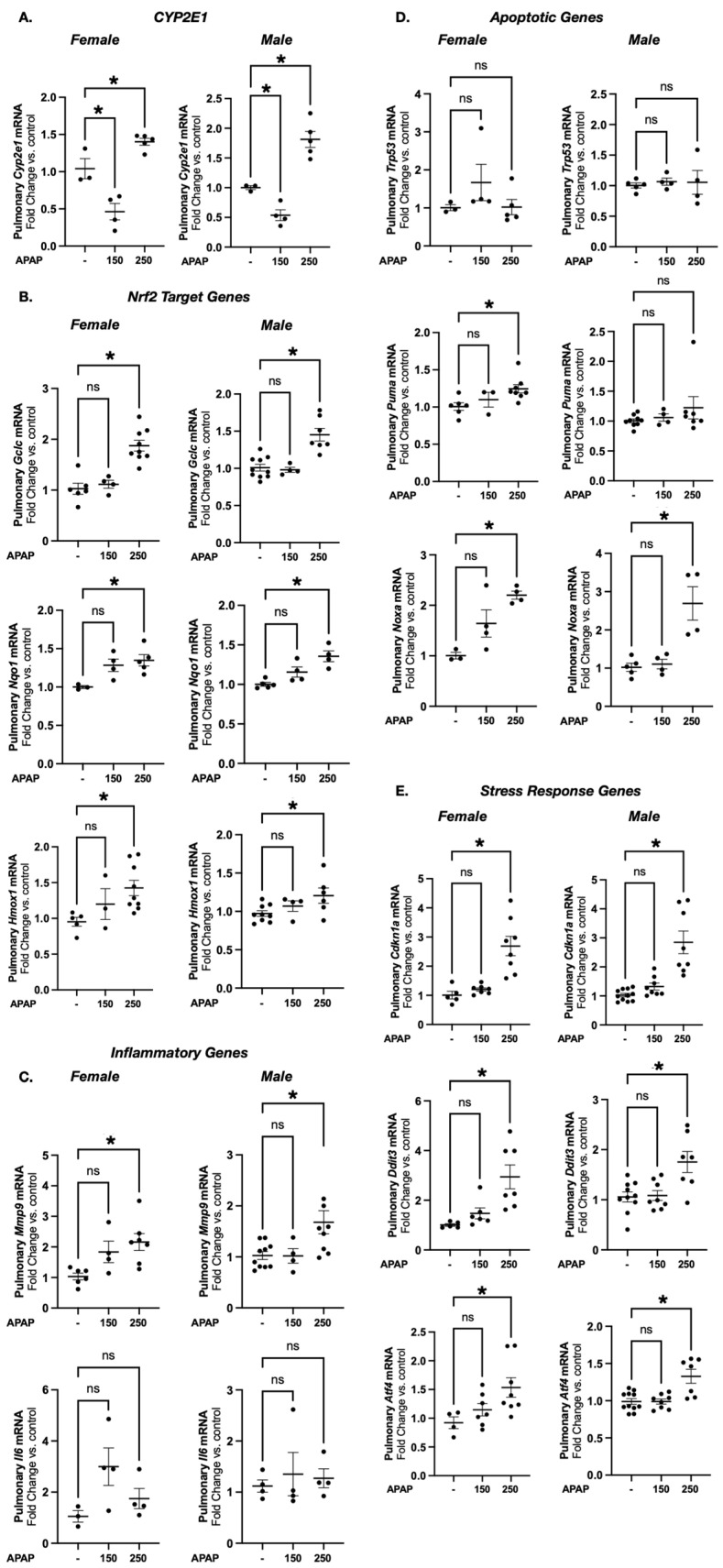
Prenatal acetaminophen exposure induces an acute transcriptional response in E18 female and male mice at 250 mg/kg. Fold change in pulmonary mRNA expression of *Cyp2e1* (**A**); NRF2 target genes *Gclc*, *Nqo1* and *Hmox1* (**B**); inflammatory genes *Mmp9* and *Il6* (**C**); apoptotic genes *Trp53*, *Puma* and *Nova* (**D**); and stress response genes *Cdkn1a*, *Ddit3* and *Atf4* (**E**) in APAP-exposed (150 or 250 mg/kg IP × 1; 6 h) female and male E17 mice. Data expressed as mean relative expression ± SEM. N = 4–9/dose. * *p* < 0.05 vs. vehicle-exposed control. ns, no statistical significance (*p* > 0.05).

**Figure 5 biomedicines-13-02446-f005:**
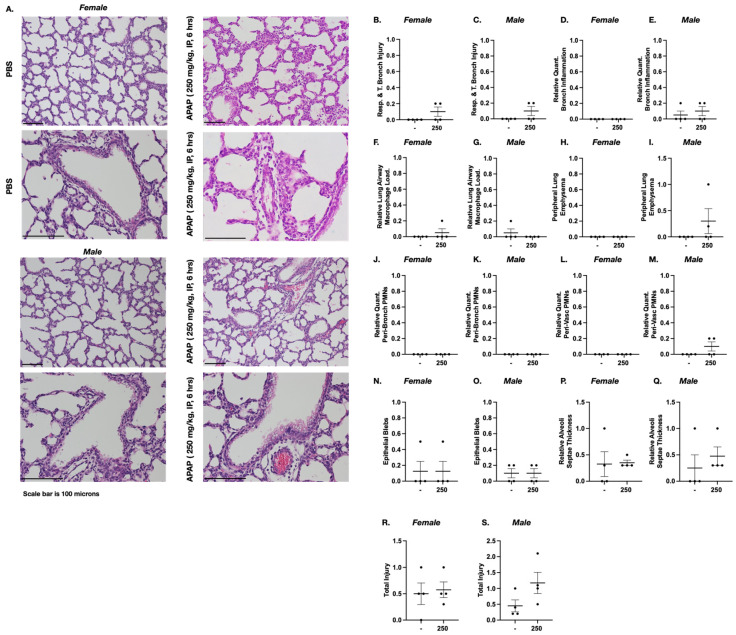
APAP-exposed fetal pulmonary tissue does not display phenotypic evidence of inflammation or injury. (**A**) Representative H&E-stained PBS and APAP (250 mg/kg IP × 1; 6 h) E18 female and male mice. (**B**–**S**) Blinded histopathological evaluation of H&E-stained pulmonary sections scored for respiratory/terminal bronchiole epithelial injury (**B**,**C**). Relative quantity bronchial-associated inflammation (**D**,**E**). Peripheral airway macrophage load (**F**,**G**). Peripheral lung emphysema (**H**,**I**). Relative quantity of peri-bronchial polymorphonuclear leukocytes (PMNs) (**J**,**K**). Relative quantity of peri-vascular PMNs (**L**,**M**). Epithelial blebs (**N**,**O**). Relative thickness of alveoli septa (**P**,**Q**). Total injury (**R**,**S**). Data expressed as mean ± SEM. N = 4/timepoint.

**Table 1 biomedicines-13-02446-t001:** List of genes and primers used for qPCR analysis.

Target	Assay ID	Protein Name (Abbreviation)
*Cype2e1*	Mm00491127_m1	Cytochrome P450, family 2, subfamily e, polypeptide 1 (CYP2E1)
*Gclc*	Mm00802655_m1	Glutamate–cysteine ligase catalytic subunit (GCLC)
*Hmox*	Mm00516005_m1	Heme oxygenase 1 (HMOX)
*Nqo1*	Mm01253561_m1	NAD(P)H dehydrogenase [quinone] 1 (NQO1)
*Mmp9*	Mm00442991_m1	Matrix metallopeptidase 9 (MMP9)
*Il6*	Mm00446190_m1	Interleukin-6 (IL-6)
*Trp53*	Mm01731290_g1	Transformation-related protein 53 (TRP53)
*Puma (Bbc3)*	Mm00519268_m1	BCL2-binding component 3 (PUMA)
*Noxa (PMAIP1)*	Mm00451763_m1	Phorbol-12-myristate-13-acetate-induced protein 1 (NOXA)
*Cdkn1a*	Mm01332263_m1	Cyclin-dependent kinase inhibitor 1A (Cdkn1a)
*Ddit3*	Mm01135937_g1	DNA damage-inducible transcript 3
*Atf4*	Mm00515325_g1	Activating Transcription Factor 4
*Sry*	Mm00441712_s1	Sex-determining region Y protein (SRY)
*18S*	Mm03928990_g1	

## Data Availability

The original contributions presented in this study are included in the article. Further inquiries can be directed to the corresponding author(s).

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
