# Peer review of "Developmentally Regulated CYP2E1 Expression Is Associated with a Fetal Pulmonary Transcriptional Response to Maternal Acetaminophen Exposure"

_biomedicines, 2025, doi:10.3390/biomedicines13102446_

Round 1

Reviewer 1 Report

Comments and Suggestions for Authors

This study investigated the effects of CYP2E1 enzyme expression at different developmental stages on the transcriptional response of fetal lungs to acetaminophen (APAP) exposure during pregnancy. Using a mouse model, the researchers found that CYP2E1 expression in the lungs peaks at embryonic day 18 (E18), and that high-dose APAP (250 mg/kg) administered to the mother at this time induces a robust transcriptional response in the fetal lungs. This response included upregulation of genes related to oxidative stress, inflammation, cell death, and cellular stress, but no histological evidence of acute lung injury was observed. This study emphasizes the critical role of developmental timing in determining fetal lung vulnerability to APAP exposure and raises the need for further investigation into long-term effects.
Regarding this, I would like to propose several points.
1. This study utilized a C57BL/6 mouse model. While animal testing is essential, significant differences may exist between rodents and humans in aspects such as lung development timing, drug metabolism processes, and physiological responses. Therefore, there are limitations to directly applying results obtained from mouse models to humans. These interspecies differences represent a significant potential limitation of the study, yet you were not explicitly mentioned. 
2. The study administered acetaminophen via intraperitoneal (IP) injection. However, humans typically take acetaminophen orally. The route of administration significantly affects a drug's absorption, distribution, metabolism, and excretion (pharmacokinetics). Therefore, intraperitoneal injection may cause different concentration changes and exposure times compared to oral administration. While these differences represent an important aspect that could reduce relevance to real-world clinical situations, you were not discussed as limitations.
3. The research team analyzed transcriptional responses by collecting tissue samples only at a single time point, 6 hours after APAP exposure. Since gene expression dynamically changes over time, results from this specific 6-hour time point alone cannot adequately represent the entire response. The peak of the response could occur earlier or later, and there is a possibility that important late-stage responses were missed.
4. Due to limitations in the analytical method, whole lung tissue was homogenized and used for gene expression analysis without isolating specific cells. While the authors cited other studies indicating that Cyp2e1 is primarily expressed in specific cells called secondary crest myofibroblasts, using whole tissue dilutes the signal from these specific cells, potentially leading to underestimation of their response or difficulty distinguishing it from responses of other cells.
5. This study focused exclusively on changes at the mRNA level (transcriptional response). Changes in mRNA expression do not always translate into changes at the protein level or alterations in actual cellular function. Since you did not measure the expression levels or activity of proteins associated with oxidative stress or cell death, or functional indicators such as glutathione depletion, you were unable to directly demonstrate whether these transcriptional changes lead to actual biological damage.

Author Response

“This study investigated the effects of CYP2E1 enzyme expression at different developmental stages on the transcriptional response of fetal lungs to acetaminophen (APAP) exposure during pregnancy. Using a mouse model, the researchers found that CYP2E1 expression in the lungs peaks at embryonic day 18 (E18), and that high-dose APAP (250 mg/kg) administered to the mother at this time induces a robust transcriptional response in the fetal lungs. This response included upregulation of genes related to oxidative stress, inflammation, cell death, and cellular stress, but no histological evidence of acute lung injury was observed. This study emphasizes the critical role of developmental timing in determining fetal lung vulnerability to APAP exposure and raises the need for further investigation into long-term effects. Regarding this, I would like to propose several points.”

We would like to thank the reviewer for spending their valuable time evaluating our work, and for this concise and complimentary summary.

C1: “This study utilized a C57BL/6 mouse model. While animal testing is essential, significant differences may exist between rodents and humans in aspects such as lung development timing, drug metabolism processes, and physiological responses. Therefore, there are limitations to directly applying results obtained from mouse models to humans. These interspecies differences represent a significant potential limitation of the study, yet you were not explicitly mentioned.”

R1: Thank you for this important point. In our original manuscript’s final paragraph of the discussion we did have the following sentence,Our data suggest that discrete developmental windows may confer heightened susceptibility to APAP-associated lung injury in mice. Whether similar windows of susceptibility exist in the developing human lungs are unknown, but if identified, such knowledge could inform clinical guidance regarding safer timing of APAP use during pregnancy.” However, we did not specifically mention the differences between mouse and human models in our limitations section. This is an important point, and we have added an explicit statement of this the discussion in the limitations section:

This study utilized C57BL/6 mouse models, recognizing significant differences that exist between rodents and humans with regards to lung development timing, drug metabolism and physiological responses. Therefore, there are limitations to directly applying results obtained from mouse models to humans.”

C2: “The study administered acetaminophen via intraperitoneal (IP) injection. However, humans typically take acetaminophen orally. The route of administration significantly affects a drug's absorption, distribution, metabolism, and excretion (pharmacokinetics). Therefore, intraperitoneal injection may cause different concentration changes and exposure times compared to oral administration. While these differences represent an important aspect that could reduce relevance to real-world clinical situations, you were not discussed as limitations.”

R2: Thank you for allowing us to improve our manuscript by drawing attention to this important point. In our work using murine models to study acetaminophen toxicity, we operate on the premise that small molecules are almost exclusively absorbed by the peritoneal vessels and delivered to the portal system. Per PMID 31873819, a limitation of IP administration is first pas liver metabolism (quoted here).

“Even though IP administration of pharmacological agents results in faster and more complete absorption compared to oral, intramuscular and SC routes, this route as any other, has certain limitations. One limitation is the first pass metabolism, similar to what is observed with orally administered drugs, because substances absorbed from the peritoneal cavity end up in portal vein and pass through the liver.”

Murine models of acetaminophen hepatic toxicity have used IP delivery almost exclusively, in part due to this reliable delivery to the liver via the portal system. However, we appreciate this reviewer bringing up this point so we can include this information in our manuscript to provide further background of our administration choice for readers. We have added this to the discussion in our section regarding the doses used in our study.

IP administration in experimental studies involving rodents has been shown to be a justifiable route for pharmacological studies due to its reliable delivery to the liver via the portal system [118]. While we know that IP administrated drugs are absorbed by peritoneal vessels and directly delivered to the portal system, one limitation is that this route does not account for small metabolic changes that may occur in the lumen of the GI tract in oral administration.” 

C3: “The research team analyzed transcriptional responses by collecting tissue samples only at a single time point, 6 hours after APAP exposure. Since gene expression dynamically changes over time, results from this specific 6-hour time point alone cannot adequately represent the entire response. The peak of the response could occur earlier or later, and there is a possibility that important late-stage responses were missed.”

R3: Thank you for allowing us to improve our manuscript by drawing attention to this point. Our original manuscript addressed how our study does not investigate the long-term impact of APAP exposure, but we did not explicitly state the need to investigate additional timepoints. We have added this important point to our discussion in the limitations section.

Additionally, this study analyzed a single time point, 6 hours post-APAP exposure. While an acute transcriptional response was observed at this time, it is likely that additional dynamic responses occur at both earlier and later stages, underscoring the need for future investigations across multiple time points.”

C4: “Due to limitations in the analytical method, whole lung tissue was homogenized and used for gene expression analysis without isolating specific cells. While the authors cited other studies indicating that Cyp2e1 is primarily expressed in specific cells called secondary crest myofibroblasts, using whole tissue dilutes the signal from these specific cells, potentially leading to underestimation of their response or difficulty distinguishing it from responses of other cells.”

R4: Thank you for this important point. While our study uses homogenized whole lung tissue rather than investigating specific cell types, we see an acute transcriptional response, indicating that there are likely even more robust cell-type specific changes that should be investigated in future studies. In our original manuscript, we comment on this necessity in the discussion section:

Future studies are needed to interrogate this hypothesis by examining cell type-specific responses to in-utero APAP exposure during critical stages of lung development.”

C5: “This study focused exclusively on changes at the mRNA level (transcriptional response). Changes in mRNA expression do not always translate into changes at the protein level or alterations in actual cellular function. Since you did not measure the expression levels or activity of proteins associated with oxidative stress or cell death, or functional indicators such as glutathione depletion, you were unable to directly demonstrate whether these transcriptional changes lead to actual biological damage.”

R5: Thank you for allowing us to improve our manuscript by drawing attention to this point. We agree that a transcriptional response does not represent actual biological damage and were cautious not to overstate our findings. In our original manuscript, we include this sentence in our discussion to emphasize this element:

“While we assessed the effects of APAP on the developing lung as a function of acute transcriptional response and histopathologic assessment, we did not examine the long-term impact of APAP exposure on lung structure or function. This is an important consideration, as a transcriptional response alone does not confirm lasting effects on lung development.”

However, we appreciate the opportunity to make this point clearer for our readers. Therefore, we have added a sentence to the abstract highlighting the need for future studies to assess protein-level changes in response to APAP exposure.

“Importantly, transcriptional responses do not directly indicate lung injury, thus future studies should assess protein-level changes following APAP exposure.”

Reviewer 2 Report

Comments and Suggestions for Authors

The manuscript of Golden et al. deals with developmentally regulated pulmonary expression of Cyp2e1 to investigate its potential association with fetal pulmonary response to maternal acetaminophen (APAP) exposure in mice of both sexes. The response included various target genes related to oxidative stress, apoptosis, inflammation, and cellular stress. The manuscript is well-structured, and the experiment was performed adequately to provide results.

Abstract

Line 22. Correct the sentence, please.

Introduction

This section is nicely written, describing literature data and the need to provide novel experimental data on maternal APAP exposure and fetal pulmonary outcome.

The Materials and Methods Section is described in detail.

Results

Lines 201- 202. Please correct the sentence.

Paragraph 3.3.2.

If I understood correctly, in the manuscript it is stated that neither dose elicited transcriptional changes in NRF2 genes or in other markers (lines 284-286), at E17. However, observing Figure 3B-E, there are significant differences in several target genes, which is also mentioned in the Discussion section in the first paragraph, as ‘’a window of transcriptional susceptibility’’ for both E17 and E18.

As a reader, for me it sounds a bit confusing when I observe Fig. 3 and see statistical differences and then read the statement that there is ‘’no change’’. I understand the emphasis is on a consistent pattern of acute transcriptional response, but when a reader observes Fig. 4, they can also see there are graphs where there is no significant change in the transcriptional response.

Did you understand what I meant by this comment? Can you somehow rephrase your sentences so there is no confusion for the reader? Maybe I am not right, but as a reader, that confused me a bit.

The Discussion section is very detailed and relevant, and contributes to the field by highlighting the need for further research.

Comments on the Quality of English Language

The English language is fine, only some sentences need correction due to technical errors.

Author Response

“The manuscript of Golden et al. deals with developmentally regulated pulmonary expression of Cyp2e1 to investigate its potential association with fetal pulmonary response to maternal acetaminophen (APAP) exposure in mice of both sexes. The response included various target genes related to oxidative stress, apoptosis, inflammation, and cellular stress. The manuscript is well-structured, and the experiment was performed adequately to provide results.”

Thank you for taking time to evaluate our manuscript in a timely manner, and for providing this concise summary of our work. It is greatly appreciated.

C6: “Abstract: Line 22. Correct the sentence, please.”

R6: Thank you for notifying us of this error. It appears we had the sentence, “We confirmed that pulmonary Cyp2e1 expression is developmentally regulated, with a significant increase at E18 in both sex sexes compared to E17” in the METHODS section of the abstract. We have removed this from the abstract as to maintain the appropriate flow.

C7: “Results: Lines 201- 202. Please correct the sentence.”

R7: Thank you for this comment. In the original manuscript submission, the sentence was, “Public murine data available from LungMAP demonstrates a prenatal surge in pulmonary Cyp2e1 at in the saccular stage of lung development [59-64].” We recognize the writing error bolded and have corrected as following:

“Public murine data available from LungMAP demonstrates a prenatal surge in pulmonary Cyp2e1 in the saccular stage lung development [59-64].”

C8: “Paragraph 3.3.2: If I understood correctly, in the manuscript it is stated that neither dose elicited transcriptional changes in NRF2 genes or in other markers (lines 284-286), at E17. However, observing Figure 3B-E, there are significant differences in several target genes, which is also mentioned in the Discussion section in the first paragraph, as ‘’a window of transcriptional susceptibility’’ for both E17 and E18.

As a reader, for me it sounds a bit confusing when I observe Fig. 3 and see statistical differences and then read the statement that there is ‘’no change’’. I understand the emphasis is on a consistent pattern of acute transcriptional response, but when a reader observes Fig. 4, they can also see there are graphs where there is no significant change in the transcriptional response.

Did you understand what I meant by this comment? Can you somehow rephrase your sentences so there is no confusion for the reader? Maybe I am not right, but as a reader, that confused me a bit.”

R8: Thank you for this important clarifying point. While there were some observed statistically significant transcriptional responses at E17, the emphasis is on a lack of consistent pattern of acute response. To clarify this, we have revised the results section for figure 3.

“While there were some statistically significant transcriptional changes observed at E17, neither dose elicited a consistent transcriptional change across NRF2 target genes or in markers of inflammation, apoptosis or cellular stress in either female or male mice at the time points investigated (Figure 3B-E).”

For E18 (figure 4) our original manuscript specifically mentions which genes had an observed significant change. We appreciate this comment bringing to our attention the confusion with these results and how they are discussed, and hope our edit brings clarification to how these results are interpreted with an emphasis on a consistent pattern of acute transcriptional response being key.

Reviewer 3 Report

Comments and Suggestions for Authors

In this murine study, the authors continued mechanistic interrogation of the association between maternal APAP and offspring pulmonary disease. The text is very well written and the methods are well described with the discussion highlighting the relevant strengths and limitations.  

Enthusiasm is mildly tempered by need for clarity in a couple of areas, as detailed below.

  1. Lines 45-49, please update with the new FDA advisory, as available here: https://www.fda.gov/news-events/press-announcements/fda-responds-evidence-possible-association-between-autism-and-acetaminophen-use-during-pregnancy
  2. Lines 453-455. Maternal illness was not reported at the lower dose of APAP in this manuscript, so it is not clear what the premise is for the sentence that maternal illness at both doses suggests that an indirect effect of maternal toxicity was not responsible for the findings in the offspring.

Author Response

“In this murine study, the authors continued mechanistic interrogation of the association between maternal APAP and offspring pulmonary disease. The text is very well written and the methods are well described with the discussion highlighting the relevant strengths and limitations.”

Thank you for taking time to evaluate our manuscript in a timely manner, and for providing this concise summary of our work. It is greatly appreciated.

C9: “Lines 45-49, please update with the new FDA advisory, as available here: https://www.fda.gov/news-events/press-announcements/fda-responds-evidence-possible-association-between-autism-and-acetaminophen-use-during-pregnancy

R9: Thank you for including this essential additional to our paper given the very recent changes to the FDAs advisory. We have included this new information in the introduction section, maintaining the history of the generally perceived safety of APAP usage during pregnancy. We have updated the references appropriately as well to include this new FDA advisory.

“Historically, there has been a general recommendation by providers that APAP use during pregnancy could be considered risk-free for both the mother and the fetus when taken at recommended doses [2, 11, 12]. Recently, the Food and Drug Administration (FDA) initiated safety label change and notified physicians of possible link between APAP use during pregnancy and subsequent diagnosis of conditions like autism and ADHD. However, they state the choice still belongs with parents and APAP remains readily available for pregnant women to use [13].”

C10: “Lines 453-455. Maternal illness was not reported at the lower dose of APAP in this manuscript, so it is not clear what the premise is for the sentence that maternal illness at both doses suggests that an indirect effect of maternal toxicity was not responsible for the findings in the offspring.”

R10: Thank you for this important point. We included this information to discuss the important consideration of maternal response to APAP at the 250 mg/kg dose as a primary driver for the observed fetal response. We have clarified this section to specify this consideration for just the higher dose of 250 mg/kg, rather than both doses used in this study, and changed “maternal illness” to “maternal response”.

It is important to note that both our study and prior reports have demonstrated hepatic injury in APAP-exposed murine dams at 250 mg/kg [67]. Thus, the fetal response we observed could in part reflect secondary effects of maternal response to APAP rather than a direct impact of APAP on the fetal lung. However, if maternal response at the 250 mg/kg dose were the primary driver, we would expect a similar impact on the embryonic lung with exposures at either E17 or E18. Further work is needed to evaluate a range of APAP doses specifically at the critical E18 timepoint.”

Round 2

Reviewer 1 Report

Comments and Suggestions for Authors

There are no further proposals to make.